# Radiotherapy Effects on Airway Management in Patients with Nasopharyngeal Cancer

**DOI:** 10.3390/cancers16223781

**Published:** 2024-11-10

**Authors:** Davut D. Uzun, Timo N. Zimmermann, Felix C. F. Schmitt, Peter K. Plinkert, Markus A. Weigand, Juergen Debus, Thomas Held, Kristin Uzun-Lang

**Affiliations:** 1Department of Anesthesiology, Medical Faculty Heidelberg, University Heidelberg, 69120 Heidelberg, Germany; deniz.uzun@med.uni-heidelberg.de (D.D.U.); felix.schmitt@med.uni-heidelberg.de (F.C.F.S.); markus.weigand@med.uni-heidelberg.de (M.A.W.); 2Department of Radiation Oncology, Medical Faculty Heidelberg, University Heidelberg, 69120 Heidelberg, Germany; timo.tryjanowski@gmail.com (T.N.Z.); juergen.debus@med.uni-heidelberg.de (J.D.); thomas.held@med.uni-heidelberg.de (T.H.); 3Department of Otorhinolaryngology, Medical Faculty Heidelberg, University Heidelberg, 69120 Heidelberg, Germany; peter.plinkert@med.uni-heidelberg.de; 4Heidelberg Institute of Radiation Oncology (HIRO), 69120 Heidelberg, Germany; 5National Center for Tumor Diseases (NCT), 69120 Heidelberg, Germany; 6Heidelberg Ion-Beam Therapy Center (HIT), Department of Radiation Oncology, Medical Faculty Heidelberg, University Heidelberg, 69120 Heidelberg, Germany; 7Clinical Cooperation Unit Radiation Oncology, German Cancer Research Center (DKFZ), 69120 Heidelberg, Germany; 8German Cancer Consortium (DKTK), Partner Site Heidelberg, 69120 Heidelberg, Germany

**Keywords:** nasopharyngeal cancer, radiotherapy, radiation toxicity, airway management, laryngoscopy, tracheal intubation, difficult tracheal intubation, videolaryngoscopy

## Abstract

The number of patients diagnosed with cancer is expected to increase in the coming years. This is attributable, among other factors, to demographic shifts in the population. It is anticipated that the number of individuals surviving cancer for an extended period will rise considerably in the future, reflecting the advances made in medical treatment. It is therefore likely that these patients will require further surgical procedures in the future, which will necessitate advanced airway management. The objective of this study is to examine the effects of radiotherapy for nasopharyngeal cancer (NPC) on the airway management performed by anesthetists. The findings of our study suggest that a notable proportion of patients with NPC will necessitate further surgical procedures throughout the course of their illness. Our study demonstrate that radiotherapy for NPC does not result in severe impairment to advanced airway management.

## 1. Introduction

The prevalence of cancer is increasing at a gradual but consistent rate, a trend that is also associated with demographic shifts in the population [1,2]. It seems probable that the number of people living with cancer and surviving for an extended period of time will increase in the future as a consequence of advances in the treatment of cancer and the implementation of early detection strategies. Therefore, it can be reasonably assumed that cancer patients will require a greater number of surgical procedures in the future, which will also necessitate the utilization of advanced airway management strategies [3].

Nasopharyngeal cancer (NPC) presents with an incidence of 84,000 and a mortality of 51,000 per year. Additionally, the incidence and mortality rates are subject to variation based on gender, ethnicity, and geography [4]. The most prevalent histological subtype is squamous cell carcinoma (SCC), with other subtypes, including basal cell carcinoma, adenocarcinoma, sarcoma, and melanoma of the mucous membranes, being observed with lower frequency. Nasopharyngeal carcinomas are classified into three distinct subgroups: Type I is designated for keratinizing tumors; Type II for non-keratinizing, differentiated tumors; and Type III for non-keratinizing, undifferentiated tumors, including lymphoepithelial histology. Given the late onset of clinical symptoms, NPC is typically diagnosed at an inoperable stage, necessitating radiation therapy as the standard treatment [5,6]. In this context, the prognosis of patients is contingent upon the type of tumor and the nodal and metastatic staging (TNM staging) as defined by the International Union Against Cancer (UICC) and the American Joint Committee on Cancer (AJCC), which classify nasopharyngeal tumors into five categories (UICC stage I-IVB) [5].

Nasopharyngeal carcinoma (NPC) is a distinct form of head and neck cancer that is thought by the scientific literature to have a genetic disposition with heterogeneous characteristics within and between tumors. Some authors posit that the nature of NPC is not an isolated genetic disease, but rather an ecological disease [7]. This should not be viewed unidimensionally, but rather as a multidimensional, spatiotemporal “unity of ecology and evolution” pathological ecosystem. The fundamental ecological principles encompass both intraspecific (e.g., communication) and interspecific (e.g., competition, predation, parasitism, and mutualism) relationships. These principles are utilized to facilitate an understanding of NPC progression [7].

For patients with UICC stage I tumors, radiotherapy (RT) is the recommended first-line treatment [8,9]. In patients with higher UICC stages (II-IVB), the concurrent administration of chemoradiotherapy has been demonstrated to reduce the risk of metastasis, enhance local control, and markedly improve overall survival (OS) rates [10,11]. The probability of locoregional failure-free survival, distant failure-free survival, and overall survival (OS) decreases with increasing UICC stage [5]. Accordingly, a combination of chemotherapy and radiotherapy is recommended for advanced stages of the disease. NPC frequently metastasizes early to the bilateral neck lymph nodes; therefore, all patients should be treated with bilateral neck RT, even if the lymph nodes are negative [12,13,14].

The introduction of advanced radiation therapy solutions has facilitated the delivery of highly targeted treatments, minimizing radiation exposure to surrounding structures and organs [15,16,17]. It should be noted, however, that RT is also associated with adverse effects [18,19]. Ionizing radiation has the potential to induce alterations in tissue, which may consequently result in modifications to the patient’s anatomy. It should be noted that these alterations could have implications for the patient’s anatomy. This is especially relevant with respect to the larynx area, where such changes could potentially result in greater complexity with regard to airway management [19]. One of the main sources of difficulty is the pathophysiological alteration that arises as a consequence of RT. During the initial phase, which typically persists for several weeks, patients may present with symptoms including dysphagia, dry mouth, mucositis, and dermatitis [20]. Additionally, edema resulting from radiotherapy can present considerable difficulties with regard to mask ventilation and advanced airway management [21]. The altered anatomy resulting from fibrosis, restricted mouth opening, and trismus can significantly impede conventional laryngoscopy, rendering the securing of the airway an exceptionally challenging endeavor [21,22].

The aim of this study is to evaluate the post-RT impact of nasopharyngeal cancer treatment on airway-related outcomes, with a particular focus on its potential as a predictor of difficult intubation, difficult bag-mask ventilation, and the influence of advanced airway management techniques, including video laryngoscopy and fiberoptic tracheal intubation.

## 2. Materials and Methods

This retrospective, single-center study was conducted based on routinely collected observational data from the clinical records of the patients. This retrospective study was conducted in accordance with the relevant institutional guidelines and in compliance with the Declaration of Helsinki of 1975, as revised in 2013. The study was approved by the local ethics committee of the medical faculty at the University of Heidelberg (reference number S-421-2015).

### 2.1. Patient Selection

The patients included in the study were selected based on a retrospective database query of the Department of Radiation Oncology at Heidelberg University Hospital, Germany. The analysis included patients aged over eighteen who had received definitive or postoperative radiation therapy for nasopharyngeal cancer at any stage between January 2012 and April 2024. The patients had undergone surgery with necessary intubation. The initial database query of all patients meeting the inclusion criteria yielded a total of 125 patients. The present study included all patients who had undergone advanced airway management, irrespective of whether the surgery took place years later or not. Patients were excluded if they had a diagnosis other than squamous cell carcinoma, had not undergone surgery at the University Hospital in Heidelberg, did not have advanced airway management for surgery, had a history of malignancies within three years before therapy, had undergone pre-irradiation, had died before surgery, or had incomplete data. The confidentiality of patients is safeguarded through the anonymization of data, which was conducted in a manner that eliminated any potential for identifying information to be discerned. Accordingly, the requirement for patient consent was waived. Following the exclusion of the aforementioned criteria, a total of 23 patients were included in the final analysis. Details are shown in Figure 1.

### 2.2. Radiotherapy Treatment Characteristics

The Department of Radiation Oncology at the University of Heidelberg was responsible for the comprehensive planning and implementation of the radiotherapy (RT) treatment. The RT was based on CT- and MRI-planned bimodal radiotherapy (RT), comprising intensity-modulated photon radiotherapy (IMRT) and carbon ion radiotherapy (CIRT), or solely photon radiotherapy as IMRT, at Heidelberg University Hospital. All patients were immobilized using bespoke thermoplastic head masks with shoulder fixation.

The Department of Otolaryngology (ENT) in Heidelberg was responsible for performing the surgical procedure following radiotherapy. The Department of Anesthesiology was tasked with administering general anesthesia and managing advanced airways for the surgical operation. Prior to the surgical procedure, the anesthesiology team conducted a comprehensive assessment of all patients during the mandatory pre-anesthesia consultation and examination [23]. Our anesthesiologists meticulously documented these findings, ensuring a detailed and accurate record of the patient’s medical history and any pertinent anesthesia-related information. The median prescribed cumulative RT-dose was 63.19 Gy (range: 50.4–74 Gy). The median single doses for the main plan were 1.8 Gy (range: 1.8–2.0 Gy), and for patients with the boost plan, the median single doses were 2.0 Gy (range: 2.0–3.0). The gross tumor volume (GTV) included all macroscopic tumor visible on the planning CT or MRI, including suspected nodal disease. In all treatment plans, the larynx was designated as an organ at risk and the dose was calculated using physical methods with the aid of the planning software.

Clinical target volume 1 (CTV1), including the macroscopic tumor or prior tumor bed in the case of surgical resection, and planning target volume 1 (PTV1), with a safety margin of 3 mm around the CTV1, were outlined for the C12 boost. Doses were prescribed on the CTV1, receiving ≥ 95% of the prescribed isodose. CTV2 included CTV1, the bilateral cervical nodal levels I–III in the case of N0, and I–IV in the case of lymphonodal spread, and, thus, corresponded at the same time to the PTV2. IMRT was conducted via Tomotherapy^®^, and IMRT doses were prescribed on CTV2, covering the CTV2 with at least 90% of the prescription dose.

A treatment plan for a patient with a T4 N2 tumor is depicted in Figure 2. The larynx is marked in yellow as an organ at risk in this plan, and the maximum laryngeal dose was 53.2 Gy. During the time period of IMRT, patients received a concomitant chemotherapeutic regime with cisplatin 40 mg/m^2^ weekly or cisplatin 100 mg/m^2^ twice.

### 2.3. Statistical Analysis

Following the extraction of data from the clinic’s internal documentation databases, a systematic recording of the study data was conducted using a database system (Microsoft Excel, Microsoft GmbH, München, Germany). Descriptive statistics were collected in detail for all data collected, in accordance with the scientific rules governing such procedures. For continuous data and results, the mean, standard deviation, minimum, median, and maximum were calculated. The statistical analyses for mean and standard deviation (SD) were conducted using IBM SPSS Statistics version 24.0.

## 3. Results

This study examined patients with NPC who had received radiotherapy and subsequently required airway management during the following surgical interventions. A total of twenty-three patients were included in the study, with the majority being male (73.9%). The mean age of the participants was 52.9 years (range: 32–85 years).

The mean Body Mass Index (BMI) was 25.7 kg/m^2^ (range: 17–35.9 kg/m^2^). Furthermore, patients exhibited relevant cardiovascular risk factors, predominantly hypertension (39.1%) and smoking (39.1%). Details of patient demographics and characteristics are shown in Table 1.

In the study cohort, 12 patients (52.2%) received treatment with bimodal radiotherapy (main course with IMRT and photons and boost course with carbon ions (C12)) and 11 patients (47.8%) received IMRT. The mean prescribed cumulative total dose for the primary tumor and lymph node metastasis was 68.2 Gy, with a mean single dose of 1.9 Gy. The mean total laryngeal dose was 53.5 Gy (range 37.67–61.1 Gy). The mean PTV volume was 918.01 ccm (range 656.3 ccm–1852.0 ccm). Detailed RT treatment characteristics are shown in Table 2. Reasons for surgery after RT in this collective were recurrent local disease (65.2%), distant metastasis (26.1%), and, in 8.7%, in case of cerebral biopsy to identify between metastasis and radiation necrosis.

Overall, 65.2% of the patients had an American Society of Anesthesiologists (ASA) class of III, followed by 30.4% ASA II (Table 3). The majority of patients had a Mallampati score of II (52.2%), followed by a Mallampati score of I (26.1%). Mallampati scores III and IV were observed in 17.3% and 4.4% of the patients. Limited mouth opening (<3 cm) was observed in 8.7% of cases. A limited range of motion of the neck was observed in 4.4% of the patients. Mask ventilation was successful in 95.5% of cases (Table 3).

Direct laryngoscopy (DL) was performed in 69.6% of the cases, followed by video laryngoscopy (VL) in 26.1%, and 4.1% required fiberoptic tracheal intubation. Overall, 47.8% of the patients had Cormack/Lehane (C/L) grade I, followed by 43.5% with C/L grade II and 8.7% with C/L grade III. In the current study, 87.0% of the patients were successfully intubated on the first attempt. No “cannot intubate, cannot oxygenate” situations occurred, and all airways could be secured without the need for Emergency Front-of-Neck Access (EFONA).

A review of complications revealed that oropharyngeal bleeding occurred in 4.3% of patients who had undergone anesthesia. There were no documented instances of severe hypoxemia, aspiration, or mortality (Table 4). The anesthetic agents employed, including neuromuscular blockade, are itemized in Table 3.

## 4. Discussion

The aim of this study is to analyze the conditions of airway management following RT in adult patients with nasopharyngeal cancer (NPC). A review of the existing literature revealed that the effects of RT on airway management can be significant [23]. However, the studies are highly heterogeneous and do not provide comprehensive coverage of all tumor entities. Furthermore, these patients frequently undergo surgical procedures in the years following RT, a course of action that is justified by the risk of complications associated with RT. Accordingly, the necessity arises for advanced airway management for upcoming surgery [24]. The prevalence of challenging airway conditions can be attributed to the potential for RT to induce substantial alterations in the patient’s anatomy [19]. In the absence of sufficient anesthesiologic preoperative evaluation, this can potentially lead to airway failure, including “can not intubation, can not oxygenate” (CICO) and even death [25].

In general, modern anesthesia is a highly safe procedure that enables a range of surgical and other invasive procedures. Despite the ongoing advancement in peri-anesthetic safety for patients, complications in the domain of airway management remain a significant concern, accounting for a considerable proportion of life-threatening complications during general anesthesia. In a recent study conducted in the United Kingdom, a cumulative 57% of perioperative cardiac arrests were attributed to airway failure and aspiration [25]. It would appear that certain patient groups are particularly susceptible to the effects of this phenomenon. It has been demonstrated that patients undergoing ear, nose, and throat (ENT) surgery and abdominal surgery, as well as those diagnosed with cancer, are at an elevated risk of experiencing complications as a result of advanced airway management [25,26]. If the airway management fails, the globally accepted method for maintaining the patient’s oxygen supply is mask ventilation. In particular, in patients with anticipated difficult airways, techniques such as apneic oxygenation can be employed to extend the duration of “safe apnoea time” [27]. Nevertheless, the edema produced by RT can also render mask ventilation considerably more challenging [20].

For example, patients undergoing RT for NPC appear to have significantly higher rates of difficult tracheal intubation (DTI) and intubation failure than the general surgical population [24]. The incidence of DTI also appears to be highly dependent on the irradiated area and proximity to the patient’s upper airway. In a recent paper, the authors demonstrated that 50% of patients exhibited challenging tracheal intubation conditions following RT [28]. In contrast with the data presented in the existing literature on difficulties encountered in advanced airway management, our study did not identify a relevant proportion of DTI cases [29,30]. In our analysis, mask ventilation was successful, easy, and without problems in 95.5% of cases. Furthermore, no oropharyngeal or nasopharyngeal airway devices were necessary to perform mask ventilation sufficiently.

The consistency of the tissue, particularly that of the soft parts, can undergo notable alterations following RT [24]. For instance, trismus represents a significant risk when performing advanced airway management. In contrast to trismus resulting from pain or inflammation, trismus caused by RT is not amenable to treatment with anesthesia or a neuromuscular blocker. In our collective of NPC patients, the radiation area included the upper airway, and the resulting toxicities can have a relevant impact. Short- and long-term toxicities after RT are known, for example, temporomandibular joint fibrosis, xerostomia, dysphagia, osteoradionecrosis, persistent edema of the upper airway tract, and aspiration pneumonia [24,31].

In a study conducted by Jayara et al., it was demonstrated that 33% of patients who underwent RT to the head and neck exhibited significantly a diminished neck range of motion, in comparison to 11.0% of patients who did not receive RT [29]. In contrast to the study by Jayara et al., our study found no evidence of restricted motion of the neck.

A recent study revealed that the incidence of unsuccessful tracheal intubation in cancer patients undergoing radiotherapy was approximately tenfold higher than that observed in the general surgical population [24]. It is important to note that these difficulties arose despite the utilization of all advanced anesthesiologic techniques, including fiberoptic tracheal intubation and video laryngoscopy (VL). In accordance with the American Society of Anesthesiologists’ (ASA) guidelines for the management of difficult airways, a tracheal intubation is deemed to have failed or to have been performed in a difficult manner when it necessitates multiple attempts [32]. In this study, the aforementioned recommendation was taken into account, and a difficult intubation was classified as requiring more than one attempt. In any discussion of the success rates of tracheal intubation or difficult conditions, it is essential to consider the multiple factors that can influence the outcome, as these can vary considerably between cases. For example, the current literature indicates that concomitant diseases also exert a significant influence. Furthermore, there is a strong correlation between obesity and an elevated physical ASA status and challenging tracheal intubation procedures. A study demonstrated that an ASA status of III or higher was an independent risk factor for difficult tracheal intubation (DTI) [33]. Furthermore, severe obesity is a contributing factor in the ASA classification. Nevertheless, the current literature is inconclusive regarding the impact of adiposity on the frequency of DTI. Some studies have indicated that adiposity is not a reliable predictor of difficult tracheal intubation [34,35]. However, other data indicate that adiposity is an independent risk factor for difficult tracheal intubation [30,36,37,38].

Poor laryngoscopy view (C/L > II) is directly associated with DTI. In a recent study, it was shown that compared to a grade I view, a statistically significant risk of DTI increased by a factor of 20.7 when compared to patients with a grade II view (OR 20.6; 95% CI 6.35–67.2, *p* < 0.0001) [24]. In our study, 91.3% of the patients had a C/L grade of I and II. Only 8.7% showed a C/L grade of III. Thus, in comparison to other studies, we could not demonstrate any relevant difficult laryngoscopy [24,28]. In principle, every effort should be made to successfully intubate the patient on the first attempt (first-pass success). There is a substantial body of evidence indicating a strong correlation between the number of attempts at tracheal intubation and adverse events [39,40]. One study demonstrated that as the number of tracheal intubation attempts increases, the incidence of adverse events, including severe hypoxemia, rises significantly [41].

The first-pass intubation success rate (FPS) in our study was 87.0%. In comparison to data from non-RT patients, this represents a relatively high success rate [42]. In a recent study, an FPS rate of 82% was reported in a mixed patient population. However, it should be noted that patients with an anticipated difficult airway were excluded from the study [42]. In a study conducted by Kriege et al., the efficacy of DL was compared with that of VL for rapid sequence intubation (RSI). Notably, the FPS rate in this study was only 71% for DL [43]. It is also pertinent to note that the success rate for tracheal intubation is significantly influenced by the level of experience of the practitioner. This is contingent upon the experience of the providers, as well as the institution in which they are employed. A recent paper from Germany demonstrated that, in anticipated difficult tracheal intubation, the use of VL was associated with a significantly superior FPS rate (97% vs. 67%) compared to DL [44]. However, it should be noted that the study should be compared cautiously, since only very experienced anesthesia consultants performed the tracheal intubations.

Our hospital is one of the largest university hospitals and cancer centers, and our anesthesiologists have considerable experience in the treatment of patients with DTI, particularly in the context of RT. Moreover, the majority of other studies exclude patients with anticipated difficult intubation conditions, which also restricts the comparability of the studies. This is also reflected in the low number of video laryngoscopy intubations; in our study, only 26.1% of patients were intubated using video laryngoscopy. In a comparable study in NPC patients, VL was used in 38.7% of cases. This is also reflected in the fact that fiberoptic awake intubation was only performed in 4.3% of cases. Furthermore, there was no airway failure or CICO situation requiring EFONA. It should be noted that a considerable number of patients were excluded from the study due to the fact that subsequent surgical procedures were not conducted at our institution, thereby precluding our ability to access the pertinent anesthesia documentation. Additionally, some patients received definitive radiotherapy and did not undergo surgery during the study period.

The anatomical area affected by radiation could also influence airway changes after RT. Scatter radiation should also be mentioned at this point, as it could play a role. However, these aspects depend heavily on the type of radiation, the irradiated volume, and the type of irradiated organ [45].

As a consequence of the growing efficacy of cancer therapy and the concomitant increase in the survival rate of cancer patients, it is anticipated that the number of individuals requiring surgical intervention and tracheal intubation following radiotherapy will also rise. The present study examined all patients with nasopharyngeal carcinoma (NPC) who underwent surgical procedures following radiotherapy (RT), including those that necessitated tracheal intubation, despite the fact that the surgery was not always directly related to the NPC. The median time interval between RT and surgical intervention in the present study was 170 weeks. Therefore, the interval between airway management and RT may also exert an influence on the anatomical pathology of the airways, and this should be taken into account in the evaluation of the studies. It is, therefore, incumbent upon anesthetists to include RT in their preoperative evaluations of patients. This is because post-RT factors such as reduced mouth opening, deformity or growth in the mouth entrance itself, reduced subluxation of the lower jaw, altered neck condition, and distorted oral cavity can render the securing of the airway more challenging [19,45].

It is important to note that our study was subject to several limitations, including its retrospective nature, the fact that it was conducted at a single center, and the relatively small number of patients included in the analysis. The number of exclusions from the study is attributable to the fact that a relevant proportion of patients were not operated on at our hospital, thereby precluding our ability to access the anesthesia documents. In the present study, there was considerable variation in the time interval between the administration of RT and the commencement of surgery, as well as in the approach to airway management. It is possible that this may have an impact on the results, and therefore, it would be beneficial to implement a standardized approach in future studies. Moreover, the efficacy of tracheal intubation is contingent upon the methodology employed and the expertise of the anesthesiologists. It should be noted that our center has considerable expertise in the treatment of patients who have undergone radiotherapy. Consequently, the results of this study may not be generalizable to physicians with different skill levels. The data from our study indicate that, at a center with our level of expertise, there is no evidence of difficult airway management in NPC patients who have undergone radiotherapy. However, the data must be interpreted with caution due to the study design and the limited number of patients.

## 5. Conclusions

The findings of this retrospective study indicate that a considerable proportion of NPC patients who have undergone radiotherapy require intervention for airway management. The effects of radiotherapy may impact perioperative care by anesthesiologists, particularly with respect to advanced airway management. While the results must be interpreted with caution, our study provides no evidence of severe impairment in advanced airway management in patients with nasopharyngeal cancer who have undergone radiotherapy.

Nevertheless, it is imperative that patients undergo a comprehensive preoperative anesthesiologic evaluation and that a bespoke strategy for advanced airway management be implemented for those who have undergone radiotherapy. This will facilitate the identification of any toxic side effects of radiotherapy and enable the effective planning of the anesthetic procedure prior to surgery, thereby enhancing perioperative patient safety.

## Figures and Tables

**Figure 1 cancers-16-03781-f001:**
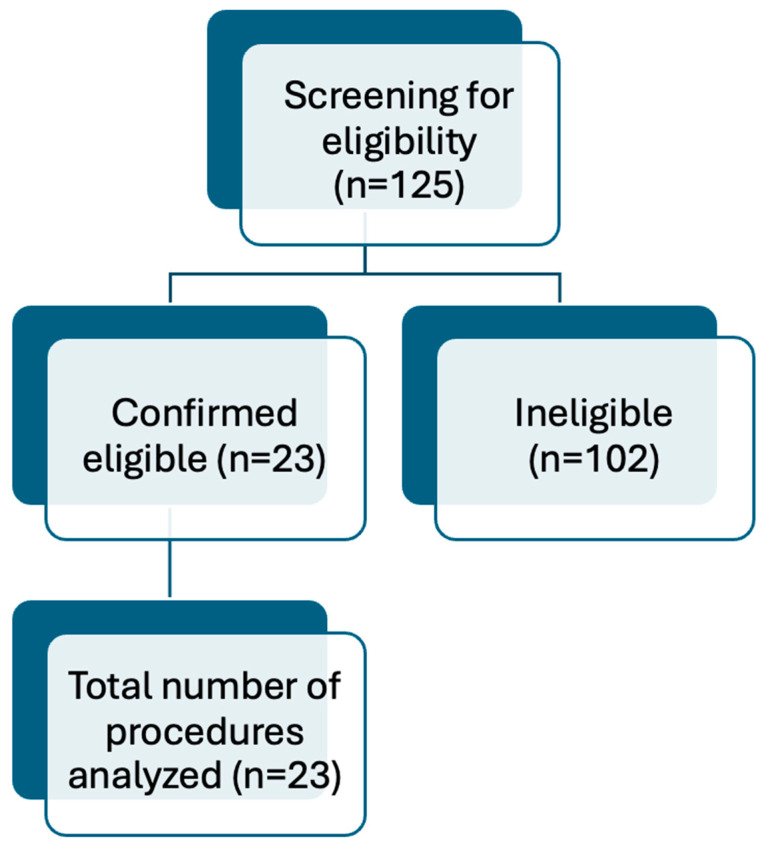
The following flowchart illustrates the number of patients who were screened, included, and excluded after undergoing radiotherapy for nasopharyngeal carcinoma.

**Figure 2 cancers-16-03781-f002:**
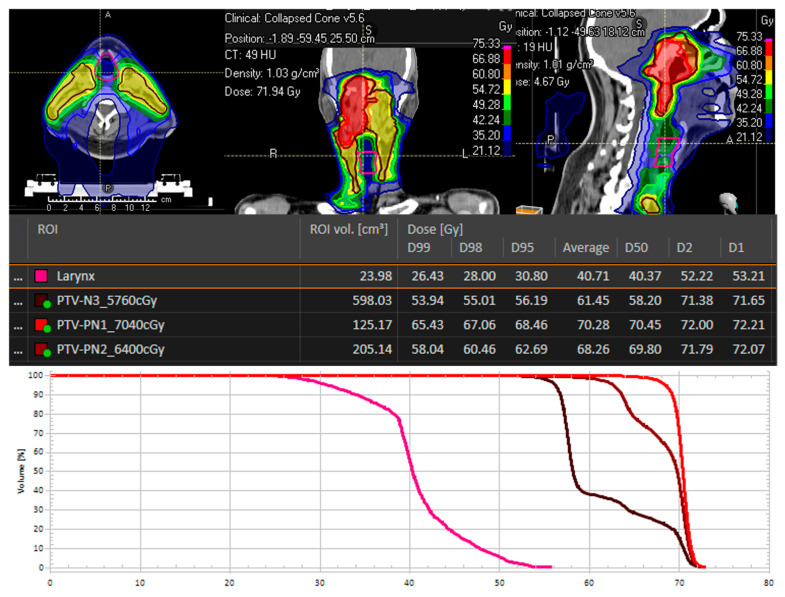
The radiation plan for a 38-year-old patient with nasopharyngeal cancer. T4 N2 M0 is presented herewith. The larynx is delineated in pink and assigned a maximum dose (D1) of 53.21 Gy.

**Table 1 cancers-16-03781-t001:** Baseline patient characteristics.

Characteristics	No. of Patients (%)
**Gender**	
Female	6 (26.1%)
Male	17 (73.9%)
**Body size**	
Mean (range)	172 cm (153–187 cm)
**Body weight**	
Mean (range)	75.8 kg (47–110 kg)
**Body Mass Index**	
Mean (range)	25.7 kg/m^2^ (17–35.9 kg/m^2^)
**Age at RT**	
Mean (range)	52.9 (32–85 years)
**Histology**	
Squamous cell carcinoma	23 (100%)
**T-stage**	
T1	5 (21.7%)
T2	4 (17.4%)
T3	4 (17.4%)
T4	10 (43.5%)
**N-stage**	
N0	4 (17.4%)
N+	19 (82.6%)
**M-stage**	
M0	23 (100%)
M1	0 (0%)
**Cardiovascular risk factors**	
Hypertension	9 (39.1%)
Pulmonary diseases	1 (4.3%)
Diabetes mellitus	2 (8.7%)
Smoking	9 (39.1%)
Coronary artery disease	2 (8.7%)
Heart failure	2 (8.7%)

**Table 2 cancers-16-03781-t002:** The characteristics of radiotherapy treatment.

Radiotherapy Treatment Characteristics	*n* (%)
**RT technique**	
Bimodal radiotherapy	12 (52.2%)
IMRT	11 (47.8%)

**Therapy regimes**	
Median total dose main plan and boost plan	68.15 Gy (range: 51.0–74.0 Gy)
Median single dose main plan and boost plan	1.9 Gy (range: 1.8–3.0 Gy)

**Irradiation cervical lymph nodes**	
Yes	23 (100.0%)
No	0 (0%)

**Mean PTV volume**	918.01 ccm (range 656.3 ccm–1852.0 ccm)
**Mean total dose laryngeal**	53.53 Gy (range 37.67–61.1 Gy)

**Dmax laryngeal**	66.61 Gy (range 62.9–68.9 Gy)
**Dmin laryngeal**	50.62 Gy (range 26.69–58.3 Gy)

**Mean time RT until surgery**	170.7 weeks (range 15–497 days)

**Reason for surgery**	
Local recurrence resection	15 (65.2%)
Metastasis resection liver, brain, distant lymph nodes	6 (26.1%)
Cerebral biopsy	2 (8.7%)

**Table 3 cancers-16-03781-t003:** Anesthesia-related characteristics.

Anesthesia Characteristics	*n* (%)
**ASA physical status**	
I (Healthy)	0 (0.0%)
II (Mild systemic illness)	7 (30.4%)
III (Severe systemic illness)	15 (65.2%)
IV (Life-threatening systemic illness)	1 (4.4%)
**Mallampati score**	
I (Soft palate, uvula, pillars visible)	6 (26.1%)
II (Soft palate, major part of uvula visible)	12 (52.2%)
III (Soft palate, base of uvula visible)	4 (17.3%)
IV (Only hard palate visible)	1 (4.4%)
**Mouth opening**	
<3 cm	2 (8.7%)
>3 cm	21 (91.3%)
**Neck range of motion**	
Full	22 (95.6%)
Limited	1 (4.4%)
**Mask ventilation**	
Easy	21 (95.5%)
Difficult	1 (4.5%)
**Intubation technique**	
Direct laryngoscopy	16 (69.6%)
Video laryngoscopy	6 (26.1%)
Fiberoptic	1 (4.3%)
Tracheostomy	0 (0.0%)
**First-pass intubation success**	
Yes	20 (87.0%)
No	3 (13.0%)
**Cormack/Lehane classification**	
I	11 (47.8%)
II	10 (43.5%)
III	2 (8.7%)
IV	0 (0.0%)
**Hypnotic drugs**	
Propofol	23 (100%)
**Analgetic drugs**	
Fentanyl	4 (17.4%)
Sufentanil	19 (82.6%)
**Neuromuscular blocking drugs**	
Rocuronium	9 (39.1%)
Atracurium	8 (34.8%)
Mivacurium	6 (26.1%)
**Anesthesia type**	
Volatile anesthetics	15 (65.2%)
Total intravenous anesthesia	8 (34.8%)

**Table 4 cancers-16-03781-t004:** Anesthesia- and airway management-related complications.

Complications During Anesthesia	*n* (%)
**Oropharyngeal bleeding**	
Yes	1 (4.3%)
No	22 (95.7%)
**Severe hypoxemia**	
Yes	0 (0.0%)
No	23 (100%)
**Aspiration**	
Yes	0 (0.0%)
No	23 (100%)
**Peri-operative mortality**	
Yes	0 (0.0%)
No	23 (100%)

## Data Availability

The authors confirm that the data supporting the findings of this study are available within the article. Because of the nature of this research, participants of this study did not agree to share their data publicly, so supporting data are not available.

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
