# Peer review of "Radiotherapy Effects on Airway Management in Patients with Nasopharyngeal Cancer"

_cancers, 2024, doi:10.3390/cancers16223781_

Round 1
Reviewer 1 Report
Comments and Suggestions for Authors
Th aim of this study is to investigate the frequency of difficult tracheal intubation in patients with NPC following RT. They also list several limitations including its retrospective nature, the fact that it was conducted at a single center, and the relatively small number of patients. In addition, some point should be noted as below,
1) The authors claim that “it is established that post-RT effects can elevate the probability of airway management complications and difficulties”, from Table 1-4, any data can show this conclusion?
2) Additionally, the title “High-Dose Radiotherapy Effects on Airway Management in Patients with Nasopharyngeal Cancer”, any data from Table 1-4 can support it? Possible not. How to define “High-Dose”, so “Low -Dose” group is ?
3) In the “Introduction” section, the advance about NPC background should be better added. A paper refreshingly proposes that NPC is an ecological disease: a multidimensional spatiotemporal "unity of ecology and evolution" pathological ecosystem (https://pubmed.ncbi.nlm.nih.gov/37056571/). This paper is suggested to be reviewed and it might provide an overall and novel perspective of NPC.
4) Abstract section “...had an ASA class of III.”, the full name of ASA should be added.
Reviewer 2 Report
Comments and Suggestions for Authors
The aim of this study was to investigate the frequency of difficult tracheal intubation in patients with NPC after RT.
The manuscript is well constructed and the message to readers is clear.
The introduction lays the foundation of the problem studied and the discussion is based on relevant literature.
Here are my comments:
- In Figure 1 the number of patients is 125, the sum of eligible and non-eligible patients 126.
- ASA when mentioned for the first time should be written in full
- In Table 3 intravenös anästhesia (mispelling)
- In contrast to the study by Jayara et al., this study found no evidence of restricted motion of the neck. Any comments to explain this difference ?
Round 2
Reviewer 1 Report
Comments and Suggestions for Authors
No other questions.